# Molecular De Novo Design through Transformer-based Reinforcement Learning

## Abstract

In this work, we introduce a method: REINVENT-Transformer to fine-tune a Transformer-based generative model for molecular de novo design. Leveraging the superior sequence learning capacity of Transformers over Recurrent Neural Networks (RNNs), our model can generate molecular structures with desired properties effectively. In contrast to the traditional RNN-based models, our proposed method exhibits superior performance in generating compounds predicted to be active against various biological targets, capturing long-term dependencies in the molecular structure sequence. The model's efficacy is demonstrated across numerous tasks, including generating analogues to a query structure and producing compounds with particular attributes, outperforming the baseline RNN-based methods. Our approach can be used for scaffold hopping, library expansion starting from a single molecule, and generating compounds with high predicted activity against biological targets.

## Introduction

The vast expanse of chemical space, encompassing an order of magnitude from $10^{60} - 10^{100}$ possible synthetically feasible molecules (Schneider & Fechner, 2005), presents formidable obstacles to drug discovery endeavors. In this colossal landscape, the task of pinpointing a molecule that simultaneously meets the prerequisites for bioactivity, drug metabolism and pharmacokinetic (DMPK) profile, and synthetic accessibility becomes an undertaking similar to the proverbial search for a needle in a haystack. Pioneering de novo design algorithms (Böhm, 1992; Gillet et al., 1994) have attempted to address this by employing virtual strategies to design and evaluate molecules, thereby condensing the vast chemical space into a more navigable realm for exploration.

Traditional de novo design models, based on Recurrent Neural Networks (RNNs), have proven effective in molecule generation tasks (Olivecrona et al., 2017). However, RNNs possess inherent architectural limitations, notably in their capability to capture long-term dependencies in sequential data, which can be particularly detrimental when modeling complex molecular structures. Recently, the Transformer architecture has emerged as a powerful alternative to RNNs in sequence modeling tasks across various domains. Some of the key advantages of Transformers over RNNs include:

1. **Parallelization:** Unlike RNNs which process sequences step-by-step, Transformers process all tokens in the sequence simultaneously, allowing for better computational efficiency.

2. **Long-term Dependency Handling:** Transformers utilize multi-head self-attention mechanisms, which can capture long-range interactions in the data, making them particularly well-suited for modeling intricate molecular structures.

3. **Scalability:** Transformers are inherently more scalable, allowing for the processing of longer sequences, which is a considerable advantage in molecular design.

In light of these advantages, our work introduces a novel approach by integrating the Transformer architecture, specifically the Decision Transformer, for molecular de novo design. By leveraging the inherent

strengths of Transformers, our model exhibits enhanced performance in generating molecular structures with desired attributes.

Furthermore, we emphasize the incorporation of the "oracle feedback reinforcement learning" method. Pre-training models on large datasets is beneficial, but downstream tasks often require fine-tuning on specific objectives. By integrating feedback from an oracle during the reinforcement learning phase, our approach can efficiently navigate the solution space, optimizing towards molecules with high predicted activity. Such oracle-guided optimization provides an added layer of precision, facilitating the generation of molecules that not only conform to structural constraints but also exhibit high bioactivity, thereby increasing the potential success rate in drug discovery endeavors.

Drawing inspiration from previous work that employed RNNs and reinforcement learning for molecular optimization (Olivecrona et al., 2017), our approach distinguishes itself by the adoption and fine-tuning of the Transformer architecture, ensuring superior handling of long-sequence data and paving the way for innovative breakthroughs in the realm of molecular design.

In summary, this work presents a fresh perspective on molecular de novo design, underscoring the potential of Transformer-based architectures, complemented by oracle feedback reinforcement learning, to revolutionize drug discovery methodologies. We envision that our approach will not only set a new benchmark in molecular generation tasks but will also inspire future research in leveraging advanced machine learning architectures for complex scientific challenges.

## Related Works

Early de novo design algorithms primarily focused on structure-based methods, where the aim was to develop ligands that precisely fit the binding pocket of a target, as highlighted in works by Böhm (Böhm, 1992) and Gillet et al. (Gillet et al., 1994). These methods, while effective in certain aspects, often resulted in molecules with suboptimal drug metabolism and pharmacokinetic (DMPK) properties, and posed challenges in synthetic tractability. In contrast, ligand-based approaches, which do not rely on the 3D structure of the target, were introduced to overcome some of these limitations. They involve creating a comprehensive virtual library of chemical structures, which are then evaluated using a scoring function (Ruddigkeit et al., 2013; Hartenfeller et al., 2012). However, as noted by Blundell et al. Blundell (1996), it is important to recognize that the effectiveness of ligand-based methods compared to structure-based ones is not definitive. Both approaches have their unique advantages and limitations, and the choice between them depends on the specific requirements and context of the drug design process.

Recently, generative models such as RNN-based methods have been used for de novo design of molecules (Segler et al., 2017; Gómez-Bombarelli et al., 2018; Yu et al., 2017). They have shown success in tasks like learning the underlying probability distribution over a large set of chemical structures, reducing the search over chemical space to only molecules seen as reasonable. Further fine-tuning of the models was done using reinforcement learning (RL) (Jaques et al., 2017), which showed considerable improvement over the initial model.

Despite these advancements, challenges such as capturing long-term dependencies in the sequence data persist. The Transformer architecture (Vaswani et al., 2023), known for its self-attention mechanism and ability to handle long sequences, has been highly successful in several sequence prediction tasks across domains. Motivated by these successes, we propose the use of Transformer-based architectures in place of RNNs for molecular de novo design.

Molecular assembly strategies, such as string-based approaches like SMILES and SELFIES (Weininger, 1988; Krenn et al., 2020), provide an efficient representation of molecules. Graph-based methods offer an intuitive two-dimensional representation of molecular structures, with nodes and edges representing atoms and bonds, respectively (Zhou et al., 2019; Jin et al., 2018). On the other hand, synthesis-based strategies aim to generate only synthesizable molecules, ensuring that the design aligns with real-world applications (Bradshaw et al., 2020; Gao et al., 2022a).

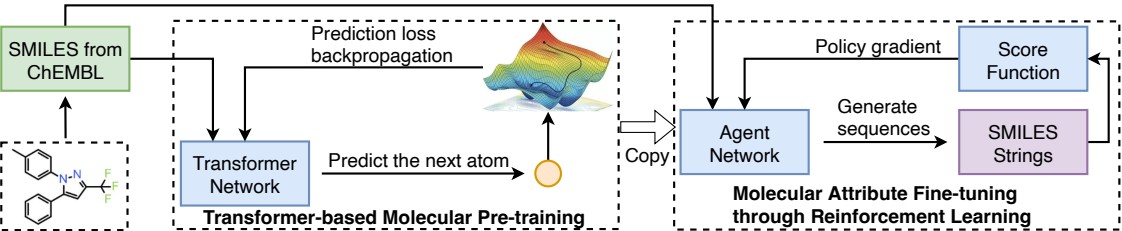

Figure 1: The framework of our method.

Various optimization algorithms have been utilized for molecular design. Genetic Algorithms (GAs) mimic natural evolutionary processes and have been applied in molecule generation using both SMILES and SELF-IES representations (Brown et al., 2019; Nigam et al., 2021). Bayesian optimization (BO) is another class of method that builds a surrogate for the objective function, with applications such as BOSS and ChemBO in the molecular domain (Moss et al., 2020; Korovina et al., 2020). Variational autoencoders (VAEs) offer a generative approach, mapping molecules to and from a latent space, with notable methods including SMILES-VAE and JT-VAE (Gómez-Bombarelli et al., 2018; Jin et al., 2018). Reinforcement Learning (RL) techniques, like REINVENT, have also been applied to tune models for molecule generation Olivecrona et al. (2017).

Furthermore, recent advancements in gradient ascent methods, such as Pasithea and Differentiable scaffolding tree (DST), have leveraged gradient-based optimization for molecular design (Shen et al., 2021; Fu et al., 2022).

The evolution of molecular design methodologies has progressively addressed various challenges and limitations. The transition from RNN-based methods to more advanced generative models underscores a quest for improved handling of complex chemical structure representations and optimization. While RNNs brought significant progress, their inherent difficulty in capturing long-term dependencies in sequential data has been a notable shortcoming. This gap is precisely where the Transformer architecture, introduced by Vaswani et al. (Vaswani et al., 2023), brings its strengths to the fore. Its self-attention mechanism allows for a more nuanced and effective handling of sequence data, a critical aspect in molecular design where long-range interactions within molecules play a pivotal role.

Parallelly, the field has seen the integration of reinforcement learning (RL) for fine-tuning generative models, as evidenced in the work by Jaques et al. (Jaques et al., 2017). RL's ability to iteratively improve models based on a feedback loop aligns well with the demands of molecular design, where continuous refinement based on molecular properties is essential. The combination of RL with generative models has been shown to enhance the ability to navigate the vast chemical space more effectively, achieving better results in molecule generation.

Therefore, in light of these advancements and limitations, we propose an approach that integrates the Transformer architecture with advanced RL techniques. This proposal is underpinned by the Transformer's superior handling of sequential data and the iterative refinement capability of RL. By merging these two powerful technologies, we aim to address the existing challenges in molecular de novo design, such as the need for better sequence representation and optimization. This integration promises to enhance the effectiveness and efficiency of molecular generation processes, moving closer to achieving more sophisticated and automated molecular design systems.

## Methodology

Our method, named as REINVENT-Transformer, first pre-trains the real 2D molecule dataset based on the transformer. Then, based on the RL paradigm, fine-tuning is performed on the molecular attributes to be optimized.

## Preliminaries

In this study, our focus is confined to versatile single-objective molecular optimization techniques that are pertinent to the design of small organic molecules. These molecules possess scalar properties that are significant in the context of therapeutic development. The molecular design challenge at hand can be formally described as an optimization task:

$$m^* = \arg \max_{m \in \mathcal{M}} \mathcal{O}(m)$$

Here, $m$ represents a molecular structure, while $\mathcal{M}$ is the expansive domain known as chemical space, encompassing all potential molecular candidates. The extent of $\mathcal{M}$ is overwhelmingly vast, for instance, around $10^{60}$ Bohacek et al. (1996). It's presupposed that we have access to the actual value of a targeted property, symbolized by $\mathcal{O}(m) : \mathcal{M} \to \mathcal{R}$, in which an oracle, $\mathcal{O}$, functions as an opaque mechanism. This oracle assesses specific chemical or biological attributes of a molecule $m$ and yields the real property $\mathcal{O}(m)$ as a scalar value. It's important to note that the oracles do not provide an analytical form or derivatives of the properties. The most feasible oracles—either experimental procedures or high-fidelity simulations—often come with considerable expense. Therefore, an algorithm that can efficiently optimize the oracle within a feasible resource allocation is crucial. Such an algorithm would be a key component in the automation of molecular design, contributing significantly to advanced automated chemical design (ACD) Goldman et al. (2022) or function-driven autonomous synthesis Gao et al. (2022b).

## Transformer-based Molecular Pre-training

The transformer is used for pre-training on real 2D molecules. Specifically, it treats the prediction of a 2D molecule as a sequence prediction and lets the transformer predict the next atom based on the molecular sequence history. The pre-training of the transformer is based on maximum likelihood.

### Training data Overview: Segmentation and Binary Coding of SMILES

A Simplified Molecular Input Line Entry System (SMILES) (Weininger, 2017) defines a molecule as a character sequence reflecting atoms as well as special symbols that illustrate ring opening and closure along with branching. In the majority of scenarios, SMILES are tokenized on a single-character basis, with the exception of two-character atom types such as " Cl " and "Br", and unique environments indicated by square brackets (e.g., [nH]), where they are processed as a single token. This approach to tokenization led to the identification of 86 tokens in the training data.

A single molecule can be represented in multiple ways using SMILES. Algorithms that consistently represent a particular molecule with the same SMILES are termed canonicalization algorithms (Weininger, 1988). Nevertheless, different algorithm implementations may still yield diverse SMILES.

**Transformers Overview**  Transformers are a neural network architecture designed to process sequential data, while also accounting for the importance of each input in relation to the others, despite their position in the sequence (Vaswani et al., 2023). They manage to do this by the introduction of an attention mechanism that assesses the significance of each input in the sequence (Figure 1). At any given step $t$, the transformer state at $t$ is influenced by all previous inputs $x_1, \ldots, x_{t-1}$ and the current input $x_t$. The transformer's ability to selectively focus on the parts of the input sequence that are most relevant for each step makes them especially well suited for tasks in the field of natural language processing. Sequences of words can be encoded into one-hot vectors with a length equivalent to our vocabulary size $X$. We may add two extra tokens, GO and EOS, to signify the beginning and end of a sequence, respectively.

A transformer block is a parameterized function class $f_\theta : \mathbb{R}^{n \times d} \to \mathbb{R}^{n \times d}$. If $\mathbf{x} \in \mathbb{R}^{n \times d}$ then $f_\theta(\mathbf{x}) = \mathbf{z}$ where

$$Q^{(h)}(\mathbf{x}_t) = W_{h,q}^T \mathbf{x}_t, \quad K^{(h)}(\mathbf{x}_t) = W_{h,k}^T \mathbf{x}_t, \quad V^{(h)}(\mathbf{x}_t) = W_{h,v}^T \mathbf{x}_t, \quad W_{h,q}, W_{h,k}, W_{h,v} \in \mathbb{R}^{d \times k} \tag{1}$$

$$\alpha_{t,j}^{(h)} = \text{softmax}_j \left( \frac{\langle Q^{(h)}(\mathbf{x}_t), K^{(h)}(\mathbf{x}_j) \rangle}{\sqrt{k}} \right), \quad \text{for } j = 1, \ldots, t \tag{2}$$

$$\mathbf{u}_t' = \sum_{h=1}^{H} W_{c,h}^T \sum_{j=1}^{t} \alpha_{t,j}^{(h)} V^{(h)}\left(\mathbf{x}_j\right), \qquad W_{c,h} \in \mathbb{R}^{k \times d} \tag{3}$$

$$\mathbf{u}_t = \mathrm{LayerNorm}\left(\mathbf{x}_t + \mathbf{u}_t'; \gamma_1, \beta_1\right), \qquad \gamma_1, \beta_1 \in \mathbb{R}^d \tag{4}$$

$$\mathbf{z}_t' = W_2^T \,\mathrm{ReLU}\left(W_1^T \mathbf{u}_t\right), \qquad W_1 \in \mathbb{R}^{d \times m}, W_2 \in \mathbb{R}^{m \times d} \tag{5}$$

$$\mathbf{z}_t = \mathrm{LayerNorm}\left(\mathbf{u}_t + \mathbf{z}_t'; \gamma_2, \beta_2\right), \qquad \gamma_2, \beta_2 \in \mathbb{R}^d \tag{6}$$

$$\hat{\mathbf{y}} = \mathrm{softmax}\left(W_z^T \mathbf{z}\right) = \frac{\exp\left(W_z^T \mathbf{z}\right)}{\sum_{k=1}^{m} \exp\left(W_z^T \mathbf{z}\right)_k}, \qquad W_z \in \mathbb{R}^{d \times o}. \tag{7}$$

The notation $\mathrm{softmax}_j$ indicates we take the softmax (defined in Equation 7) over the $d$-dimensional vector indexed by $j$. The LayerNorm function Ba et al. (2016) is defined for $\mathbf{z} \in \mathbb{R}^k$ by

$$\mathrm{LayerNorm}(\mathbf{z}; \gamma, \beta) = \gamma \frac{(\mathbf{z} - \mu_{\mathbf{z}})}{\sigma_{\mathbf{z}}} + \beta, \quad \gamma, \beta \in \mathbb{R}^k \tag{8}$$

$$\mu_{\mathbf{z}} = \frac{1}{k} \sum_{i=1}^{k} \mathbf{z}_i, \quad \sigma_{\mathbf{z}} = \sqrt{\frac{1}{k} \sum_{i=1}^{k} \left(\mathbf{z}_i - \mu_{\mathbf{z}}\right)^2} \tag{9}$$

The set of parameters, denoted by $\theta$, comprises the elements of the weight matrices $W$ and the LayerNorm parameters $\gamma$ and $\beta$, as specified on the right-hand side. The input $\mathbf{x} \in \mathbb{R}^{n \times d}$ represents a set of $n$ entities, each characterized by $d$ attributes (typically, though not exclusively, sequences of $d$-dimensional vectors of length $n$). It is important to note that the output $\mathbf{z} \in \mathbb{R}^{n \times d}$ retains the same format as the input $\mathbf{x} \in \mathbb{R}^{n \times d}$. A transformer is an amalgamation of $L$ distinct transformer blocks, each equipped with unique parameters: $f_{\theta_L} \circ \cdots \circ f_{\theta_1}(\mathbf{x}) \in \mathbb{R}^{n \times d}$. Key hyperparameters in a transformer include $d, k, m, H$, and $L$, with typical configurations being $d = 512, k = 64, m = 2048, H = 8$. While the initial research suggested $L = 6$, more recent studies tend to employ a greater number of these blocks.

**Learning to model the data** Training a Transformer for sequence modeling typically involves using maximum likelihood estimation to predict the next token $x_t$ in the target sequence, given tokens from the previous steps (Figure 1). The model generates a probability distribution at every step, representing the likely next character, and the objective is to maximize the likelihood assigned to the correct token:

$$J(\Theta) = -\sum_{t=1}^{T} \log P\left(x_t \mid x_{t-1}, \ldots, x_1\right) \tag{10}$$

The cost function $J(\Theta)$, often applied to a subset of all training examples known as a batch, is minimized with respect to the network parameters $\Theta$. Given a predicted log likelihood $\log P$ of the target at step $t$, the gradient of the cost function with respect to $\Theta$ is used to update $\Theta$. This method of fitting a neural network is called back-propagation. Changing the network parameters affects not only the immediate output at time $t$, but also influences the information flow into subsequent transformer states.

**Generating new samples** Once a Transformer has been trained on target sequences, it can be used to generate new sequences that adhere to the conditional probability distributions learned from the training set. The first input is the GO token, and at every timestep following, we sample an output token $x_t$ from the predicted probability distribution $P(X_t)$ over our vocabulary $X$. The sampled $x_t$ is then used as our next input. The sequence is considered finished once the EOS token is sampled .

---

**Algorithm 1** REINVENT Transformer Pretraining Process

---

 1: **Function Pretrain(restore_from=None):**
 2: Initialize Vocabulary from file
 3: Load and preprocess data from 'ZINC' and 'ChEMBL'
 4: Filter and prepare the dataset
 5: Create a DataLoader for batch processing
 6: Initialize the Transformer model
 7: **if** restore_from is not None **then**
 8:     Load saved model state
 9: **end if**
10: Initialize optimizer with learning rate
11: **for** each epoch **do**
12:     **for** each batch in DataLoader **do**
13:         Sample sequences (seqs) from DataLoader
14:         Compute log probability (log_p) with Transformer model
15:         Calculate loss: $loss = -\text{mean}(\log\_p)$
16:         Zero gradients
17:         Perform backpropagation
18:         Update model parameters
19:         **if** step % adjustment_interval == 0 **then**
20:             Decrease learning rate by a specified factor
21:             Sample a set of sequences for validation
22:             Decode sampled sequences to SMILES
23:             Validate the chemical structure of each SMILES
24:             Calculate the percentage of valid SMILES
25:             Display current epoch, step, loss, and % valid SMILES
26:         **end if**
27:     **end for**
28:     Save the current state of the Transformer model
29: **end for**
30: **End Function**
31: Call Pretrain function

---

---

**Algorithm 2** REINVENT Transformer Optimization Process

---

1: **Initialization**
2: Prior, Agent ← Transformer(Vocabulary)
3: Optimizer ← Adam(Agent.parameters, lr=config['learning_rate'])
4: Experience ← ExperienceReplay(Vocabulary)
    **Training Loop**
5: **while** True **do**
6:   **if** len(oracle) > 100 **then**
7:     Sort oracle buffer
8:     old_scores ← first 100 scores from oracle buffer
9:   **else**
10:     old_scores ← 0
11:   **end if**
    **Sampling and Evaluating Sequences**
12:   Seqs, AgentLikelihood, Entropy ← Agent.sample(config['batch_size'])
13:   UniqueIdxs ← Unique(Seqs)
14:   Seqs,AgentLikelihood,Entropy ←Seqs[UniqueIdxs],AgentLikelihood[UniqueIdxs],Entropy[UniqueIdxs]

15:   PriorLikelihood, - ← Prior.likelihood(Seqs)
16:   SMILES ← seq_to_smiles(Seqs, Vocabulary)
17:   Score ← Oracle(SMILES)
18:   **if** finish condition met **then**
19:     Break loop
20:   **end if**
21:   **if** len(oracle) > 1000 **then**
22:     Check for convergence based on new scores and old scores
23:     **if** convergence criteria met **then**
24:       Break loop
25:     **end if**
26:   **end if**
    **Loss Calculation**
27:   AugmentedLikelihood ← PriorLikelihood.float() + config['sigma'] × Score.float()
28:   Loss ← mean((AugmentedLikelihood - AgentLikelihood)$\hat{}$2)
    **Experience Replay (if enabled)**
29:   **if** config['experience_replay'] and len(Experience) > config['experience_replay'] **then**
30:     Experience replay steps
31:   **end if**
    **Optimization**
32:   Update experience with new experience
33:   LossRegularizer ← -mean(1 / AgentLikelihood)
34:   TotalLoss ← Loss + 5 × 10$\hat{}$3 × LossRegularizer
35:   Optimizer.zero_grad()
36:   TotalLoss.backward()
37:   Optimizer.step()
38:   Increment step counter
39: **end while**

---

## Molecular Attribute Fine-tuning through Reinforcement Learning

In this part, we load the pre-trained transformer network and fine-tune it based on RL. Here, our task is to generate some specific molecules with good attributes. Therefore, we use the generated molecules to measure the properties of the corresponding molecules through Oracle, and use them as rewards to finetune the neural network.

**Agent Decision-Making and Markov Decision Processes**  Assume an Agent that must decide on an action $a \in \mathbb{A}(s)$ to take given a particular state $s \in \mathbb{S}$, where $\mathbb{S}$ denotes the set of possible states and $\mathbb{A}(s)$ represents the set of potential actions for that state. The policy $\pi(a \mid s)$ of an Agent associates a state to the likelihood of each action executed within. Reinforcement learning challenges are often depicted as Markov decision processes, indicating that the current state provides all essential information to inform our action choice, and no additional benefit is gained from knowing past states' history. While this is more of an approximation than a fact for most real-life challenges, we can extend this concept to a partially observable Markov decision process where the Agent interacts with a partial environment representation. Let $r(a \mid s)$ be the reward serving as an indicator of the effectiveness of an action taken at a certain state, and the long-term return $G(a_t, S_t) = \sum_t^T r_t$. represents the cumulative rewards collected from time $t$ to time $T$ (Sutton & Barto, 1999). As molecular desirability is only meaningful for a completed SMILES, we will only consider a complete sequence's return.

The main objective of reinforcement learning is to enhance the Agent's policy to increase the expected return $\mathbb{E}[G]$ based on a set of actions taken from some states and the obtained rewards. A task with a definitive endpoint at step $T$ is known as an episodic task (Sutton & Barto, 1999), where $T$ corresponds to the episode's length. SMILES generation is an example of an episodic task, which concludes once the EOS token is sampled.

The states and actions used for Agent training can be produced by the agent itself or through other means. If the agent generates them, the learning is called on-policy, and if generated by other means, it is off-policy learning (Sutton & Barto, 1999).

Reinforcement learning commonly employs two different strategies to determine a policy: value-based RL and policy-based RL (Sutton & Barto, 1999). In value-based RL, the aim is to learn a value function that describes a given state's expected return. Once this function is learned, a policy can be established to maximize a certain action's expected state value. In contrast, policy-based RL aims to learn a policy directly. For the problem we are addressing, we believe policy-based methods are the most suitable for the following reasons:

- Policy-based methods can explicitly learn an optimal stochastic policy (Sutton & Barto, 1999), which aligns with our objective.

- The used method starts with a prior sequence model. The goal is to fine-tune this model based on a specific scoring function. Since the prior model already embodies a policy, fine-tuning might require only minimal changes to the prior model. The short and fast-sampling episodes in this case decrease the gradient estimate's variance impact.

**Negative Log-Likelihood (NLL) and Loss Function**  To assess the likelihood of sequence generation by the agent, we use the Negative Log-Likelihood (NLL). The NLL is calculated as follows:

$$NLL(S) = -\sum_{i=1}^{N} \ln P\left(X_i = T_i \mid X_{i-1} = T_{i-1} \ldots X_1 = x_1\right) \tag{11}$$

This measure is crucial in understanding the generative model's performance (Blaschke et al., 2020). The augmented likelihood and loss function are then computed to adjust the agent's generation process:

$$NLL(\boldsymbol{S})_{\text{Augmented}} = NLL(\boldsymbol{S})_{\text{Prior}} - \boldsymbol{\sigma} * MPO(\boldsymbol{S})_{\text{score}}$$
$$\text{loss} = \left[ NLL(\boldsymbol{S})_{\text{Augmented}} - NLL(\boldsymbol{S})_{\text{Agent}} \right]^2$$

**Scoring Functions for Molecular Sequences** REINVENT-Transformer utilizes scoring functions to evaluate and guide the generation of molecular sequences. These functions are formulated as either a weighted product or a weighted sum:

$$S(x) = \left[ \prod_i p_i(x)^{w_i} \right]^{1 / \sum_i w_i}$$
$$S(x) = \frac{\sum_i w_i * p_i(x)}{\sum_i w_i}$$

This scoring approach is designed to balance various molecular properties during the generation process, facilitating the production of molecules with desired characteristics (Blaschke et al., 2020).

## Experiment

### Dataset

For any method that necessitates a database, we exclusively use the ZINC 250K dataset Irwin & Shoichet (2005). This dataset comprises approximately 250K molecules, selected from the ZINC database due to their pharmaceutical significance, manageable size, and widespread recognition. Both Screening Ahmed et al. (2018) and MolPAL Graff et al. (2021) conduct searches within this database. Additionally, generative models like VAEs Kingma & Welling (2014) and LSTMs Yu et al. (2019) are pretrained on it. Any fragments essential for JT-VAE (Jin et al., 2018), MIMOSA (Fu et al., 2021), and DST (Fu et al., 2022) are also derived from this very database.

### Baseline

To make a comprehensive comparison, eight baseline methods are adopted in performance evaluation.

First, we compare two rule-based baselines, shown in the Table 1

### Metric

In order to evaluate both optimization capability and sample efficiency, we utilize the area under the curve (AUC) of the top-$K$ average property value in relation to the number of oracle calls. This metric, which we refer to as *AUC top-K*, is formally defined as follows:

Given a sequence of molecules $\{M_1, M_2, \ldots, M_N\}$ generated by a method, and an oracle function $O(M)$ that returns the property value of a molecule, the top-$K$ average property value at any point in the sequence is given by:

$$\text{Top-K Average}(M_1, M_2, \ldots, M_i) = \frac{1}{K} \sum_{j=1}^{K} O(M_{(j)}) \tag{12}$$

where $M_{(j)}$ is the $j$-th highest property value molecule among the first $i$ molecules.

The *AUC top-K* is then the area under the curve when plotting the top-$K$ average property value against the number of oracle calls up to molecule $M_i$, for $i = 1$ to $N$. This is calculated as:

$$\text{AUC top-K} = \int_1^N \text{Top-K Average}(M_1, M_2, \ldots, M_i) \, di \tag{13}$$

| Method | Overview | Technical Details | Advantage | Disadvantage |
|---|---|---|---|---|
| REINVENT | A method employing a policy-based reinforcement learning approach to instruct RNNs to produce SMILES strings. | Formulates molecular design as a Markov decision process with states representing partially generated molecules and actions as string manipulations. Rewards based on properties of interest. | Adaptable to generate other string representations like SELFIES. | Heavily reliant on the design of rewards. |
| Graph-GA | A genetic algorithm that manipulates molecular representations using graphs, with graph matching and atom/fragment mutations. | Introduces crossover operations based on graph representations, unlike string-based genetic algorithms. | Offers a richer set of operations for exploring diverse chemical spaces. | Increased complexity due to graph-based operations. |
| SELFIES-REINVENT | An extension of REINVENT for generating SELF-referencing Embedded Strings (SELFIES). | Uses a policy-based RL approach for SELFIES representation, ensuring syntactical validity. | Produces molecules with fewer syntactical errors. | Still dependent on reward system definition. |
| GP BO | Combines Gaussian process Bayesian optimization with Graph-GA methods. | Leverages GP acquisition function integrated with Graph-GA techniques for sampling. | Balances exploration and exploitation effectively. | Higher computational costs due to GP and GA interplay. |
| STONED | A modified genetic algorithm that manipulates tokens within SELFIES strings. | Interacts directly with tokens in SELFIES strings, differing from traditional string-based GAs. | Direct approach potentially reduces invalid chemical representations. | Limited to SELFIES, may not generalize to other representations. |
| SMILES-LSTM HC | Iterative learning method using LSTM to understand molecular distribution in SMILES strings. | Employs a variant of the cross-entropy method, fine-tuning the model with high-scoring molecules. | Iteratively refines the generative process. | Slow convergence if initial model is suboptimal. |
| SMILES-GA | Genetic algorithm based on SMILES context-free grammar. | Implements genetic mutations and crossovers based on SMILES grammar. | Exploits SMILES structure for effective exploration. | Confined to SMILES grammar nuances, potentially missing novel structures. |
| SynNet | Synthesis-based genetic algorithm operating on binary fingerprints and decoding to synthetic pathways. | Focuses on synthesizability of generated molecules. | Prioritizes synthesizability, ensuring lab producibility of molecules. | Limited diversity in molecular space exploration due to synthesis emphasis. |
| DoG-Gen | Tailored to learn the distribution of synthetic pathways. | Represents synthetic pathways as DAGs, using an RNN generator for modeling. Emphasizes synthesizability. | Structured approach to learning synthetic pathways. | Issues in capturing very long sequences with RNNs if not designed effectively. |
| DST | Differentiable Scaffolding Tree method for molecular optimization using gradient ascent. | Abstracts molecular graphs into scaffolding trees, using a graph neural network for gradient estimation. | Direct optimization of molecular structures through gradient computation. | Possible loss of information due to abstraction to scaffolding trees. |

Table 1: Summary of Methods in Molecular Design

We set $K$ at 1, 10, and 100, capping the number of oracle calls at 10,000. All AUC values reported are min-max scaled to the range [0, 1].

**Recall (Sensitivity):** Traditionally, recall is the proportion of actual positives correctly identified. In our context, it is the proportion of molecules with desirable properties (as judged by the oracle) that the method successfully identifies from the total 'N' molecules deemed desirable by the oracle.

**Precision (Positive Predictive Value):** Precision is the proportion of predicted positives that are true positives. Here, it is the proportion of molecules identified by the method as having desirable properties that are indeed validated by the oracle, out of the 'M' molecules selected by the method.

| Method | REINVENT-Trans | REINVENT | Graph GA | REINVENT | GP BO | STONED |
| Assembly | SMILES | SMILES | Fragments | SELFIES | Fragments | SELFIES |
|---|---|---|---|---|---|---|
| Albuterol_Similarity | **0.910± 0.008** | 0.882± 0.006 | 0.838± 0.016 | 0.826± 0.030 | 0.898± 0.014 | 0.745± 0.076 |
| Amlodipine_MPO | 0.653± 0.029 | 0.635± 0.035 | **0.661± 0.020** | 0.607± 0.014 | 0.583± 0.044 | 0.608± 0.046 |
| Celecoxib_Rediscovery | 0.457± 0.071 | 0.713± 0.067 | 0.630± 0.097 | 0.573± 0.043 | **0.723± 0.053** | 0.382± 0.041 |
| DRD2 | 0.931± 0.006 | 0.945± 0.007 | 0.964± 0.012 | 0.943± 0.005 | 0.923± 0.017 | 0.913± 0.020 |
| Deco_Hop | 0.645± 0.038 | 0.666± 0.044 | 0.619± 0.004 | 0.631± 0.012 | 0.629± 0.018 | 0.611± 0.008 |
| Fexofenadine_MPO | 0.796± 0.007 | 0.784± 0.006 | 0.760± 0.011 | 0.741± 0.002 | 0.722± 0.005 | **0.797± 0.016** |
| Isomers_C9H10N2O2PF2Cl | 0.809± 0.040 | 0.642± 0.054 | 0.719± 0.047 | 0.733± 0.029 | 0.469± 0.180 | 0.805± 0.031 |
| Median 1 | 0.354± 0.008 | **0.356± 0.009** | 0.294± 0.021 | 0.355± 0.011 | 0.301± 0.014 | 0.266± 0.016 |
| Median 2 | 0.263± 0.006 | 0.276± 0.008 | 0.273± 0.009 | 0.255± 0.005 | **0.297± 0.009** | 0.245± 0.032 |
| Mestranol_Similarity | **0.685± 0.032** | 0.618± 0.048 | 0.579± 0.022 | 0.620± 0.029 | 0.627± 0.089 | 0.609± 0.101 |
| Osimertinib_MPO | 0.813± 0.010 | **0.837± 0.009** | 0.831± 0.005 | 0.820± 0.003 | 0.787± 0.006 | 0.822± 0.012 |
| Perindopril_MPO | 0.525± 0.011 | 0.537± 0.016 | 0.538± 0.009 | 0.517± 0.021 | 0.493± 0.011 | 0.488± 0.011 |
| QED | **0.942± 0.000** | 0.941± 0.000 | 0.940± 0.000 | 0.940± 0.000 | 0.937± 0.000 | 0.941± 0.000 |
| Ranolazine_MPO | 0.761± 0.012 | 0.742± 0.009 | 0.728± 0.012 | 0.748± 0.018 | 0.735± 0.013 | **0.765± 0.029** |
| Scaffold_Hop | **0.560± 0.013** | 0.536± 0.019 | 0.517± 0.007 | 0.525± 0.013 | 0.548± 0.019 | 0.521± 0.034 |
| Sitagliptin_MPO | **0.563± 0.025** | 0.451± 0.003 | 0.433± 0.075 | 0.194± 0.121 | 0.186± 0.055 | 0.393± 0.083 |
| Thiothixene_Rediscovery | 0.556± 0.016 | 0.534± 0.013 | 0.479± 0.025 | 0.495± 0.040 | **0.559± 0.027** | 0.367± 0.027 |
| Troglitazone_Rediscovery | **0.451± 0.015** | 0.441± 0.013 | 0.390± 0.016 | 0.348± 0.012 | 0.410± 0.015 | 0.320± 0.018 |
| Valsartan_Smarts | **0.165± 0.278** | 0.165± 0.358 | 0.000± 0.000 | 0.000± 0.000 | 0.000± 0.000 | 0.000± 0.000 |
| Zaleplon_MPO | **0.544 ± 0.041** | 0.358± 0.062 | 0.346± 0.032 | 0.333± 0.026 | 0.221± 0.072 | 0.325± 0.027 |
| sum | 12.197 | 12.047 | 11.526 | 11.092 | 11.152 | 10.598 |
| rank | 1 | 2 | 3 | 4 | 5 | 6 |

| Method | LSTM HC | SMILES GA | SynNet | DoG-Gen | DST | |
| Assembly | SMILES | SMILES | Synthesis | Synthesis | Fragments | |
|---|---|---|---|---|---|---|
| Albuterol_similarity | 0.719± 0.018 | 0.661± 0.066 | 0.584± 0.039 | 0.676± 0.013 | 0.619± 0.020 | |
| Amlodipine_MPO | 0.593± 0.016 | 0.549± 0.009 | 0.565± 0.007 | 0.536± 0.003 | 0.516± 0.007 | |
| Celecoxib_Rediscovery | 0.539± 0.018 | 0.344± 0.027 | 0.441± 0.027 | 0.464± 0.009 | 0.380± 0.006 | |
| DRD2 | 0.919± 0.015 | 0.908± 0.019 | **0.969± 0.004** | 0.948± 0.001 | 0.820± 0.014 | |
| Deco_Hop | **0.826± 0.017** | 0.611± 0.006 | 0.613± 0.009 | 0.800± 0.007 | 0.608± 0.008 | |
| Fexofenadine_MPO | 0.725± 0.003 | 0.721± 0.015 | 0.761± 0.015 | 0.695± 0.003 | 0.725± 0.005 | |
| Isomers_C9H10N2O2PF2Cl | 0.342± 0.027 | **0.860± 0.065** | 0.241± 0.064 | 0.199± 0.016 | 0.458± 0.063 | |
| Median 1 | 0.255± 0.010 | 0.192± 0.012 | 0.218± 0.008 | 0.217± 0.001 | 0.232± 0.009 | |
| Median 2 | 0.248± 0.008 | 0.198± 0.005 | 0.235± 0.006 | 0.212± 0.000 | 0.185± 0.020 | |
| Mestranol_Similarity | 0.526± 0.032 | 0.469± 0.029 | 0.399± 0.021 | 0.437± 0.007 | 0.450± 0.027 | |
| Osimertinib_MPO | 0.796± 0.002 | 0.817± 0.011 | 0.796± 0.003 | 0.774± 0.002 | 0.785± 0.004 | |
| Perindopril_MPO | 0.489± 0.007 | 0.447± 0.013 | **0.557± 0.011** | 0.474± 0.002 | 0.462± 0.008 | |
| QED | 0.939± 0.000 | 0.940± 0.000 | 0.941± 0.000 | 0.934± 0.000 | 0.938± 0.000 | |
| Ranolazine_MPO | 0.714± 0.008 | 0.699± 0.026 | 0.741± 0.010 | 0.711± 0.006 | 0.632± 0.054 | |
| Scaffold_Hop | 0.533± 0.012 | 0.494± 0.011 | 0.502± 0.012 | 0.515± 0.005 | 0.497± 0.004 | |
| Sitagliptin_MPO | 0.066± 0.019 | 0.363± 0.057 | 0.025± 0.014 | 0.048± 0.008 | 0.075± 0.032 | |
| Thiothixene_Rediscovery | 0.438± 0.008 | 0.315± 0.017 | 0.401± 0.019 | 0.375± 0.004 | 0.366± 0.006 | |
| Troglitazone_Rediscovery | 0.354± 0.016 | 0.263± 0.024 | 0.283± 0.008 | 0.416± 0.019 | 0.279± 0.019 | |
| Valsartan_Smarts | 0.000± 0.000 | 0.000± 0.000 | 0.000± 0.000 | 0.000± 0.000 | 0.000± 0.000 | |
| Zaleplon_MPO | 0.206± 0.006 | 0.334± 0.041 | 0.341± 0.011 | 0.123± 0.016 | 0.176± 0.045 | |
| sum | 10.227 | 10.185 | 9.613 | 9.554 | 9.203 | |
| rank | 7 | 8 | 9 | 10 | 11 | |

Table 2: Performance comparison between REINVENT-Transformer, REINVENT, and other methods over all oracles for AUC Top-10

## Evaluation Results

Our result is shown in Table. 2. From the table, we can observe that our method is better than the baseline method on multiple Oracles, which proves the effectiveness of the transformer in our problem.

### Overall Molecular Generation Result

The evaluation results depict a thorough comparison between the REINVENT-Transformer (referred to as REINVENT-Trans) and other prominent models across multiple oracles. Randomly selected SMILES generated by different models can be seen in Table 4. And the corresponding chemical structures are shown in figure 2

### Overall Molecular Generation Result Performance Overview

**REINVENT-Transformer** demonstrates its strength in molecular generation, consistently achieving top results in several properties. For instance, the model achieved the highest performance for 'Albuterol_Similarity', 'Mestranol_Similarity', 'QED', 'Scaffold_Hop', and 'Sitagliptin_MPO'. This suggests that the transformer's architecture potentially excels in capturing intricate molecular patterns and relations, and effectively optimizing towards desired properties.

### Comparative Insight

| Oracle | Model | Avg SA↓ | Diversity Top100 ↑ |
|---|---|---|---|
| Albuterol Similarity | REINVENT | 3.177 | 0.394 |
| | REINVENT-Trans | **3.173** | **0.408** |
| Amlodipine MPO | REINVENT | **3.478** | **0.391** |
| | REINVENT-Trans | 3.888 | 0.311 |
| Celecoxib Rediscovery | REINVENT | 3.458 | **0.551** |
| | REINVENT-Trans | **3.245** | 0.357 |
| DRD2 | REINVENT | **2.788** | **0.868** |
| | REINVENT-Trans | 2.914 | 0.464 |
| Deco Hop | REINVENT | 3.458 | **0.551** |
| | REINVENT-Trans | **3.240** | 0.457 |
| Fexofenadine MPO | REINVENT | 4.163 | 0.325 |
| | REINVENT-Trans | **4.113** | **0.411** |
| GSK3B | REINVENT | **3.146** | **0.884** |
| | REINVENT-Trans | **3.146** | **0.884** |
| Isomers C7H8N2O2 | REINVENT | 4.273 | 0.712 |
| | REINVENT-Trans | **2.589** | **0.796** |
| Isomers C9H10N2O2PF2Cl | REINVENT | 3.261 | 0.585 |
| | REINVENT-Trans | **3.245** | **0.686** |
| Median 1 | REINVENT | 4.571 | **0.408** |
| | REINVENT-Trans | **3.532** | 0.371 |
| Median 2 | REINVENT | **2.772** | **0.411** |
| | REINVENT-Trans | 2.877 | 0.389 |
| Mestranol Similarity | REINVENT | **3.799** | 0.267 |
| | REINVENT-Trans | 4.394 | **0.434** |
| Osimertinib MPO | REINVENT | **3.174** | **0.504** |
| | REINVENT-Trans | 3.799 | 0.447 |
| Perindopril MPO | REINVENT | 3.819 | **0.479** |
| | REINVENT-Trans | **3.766** | 0.357 |
| QED | REINVENT | **1.883** | **0.573** |
| | REINVENT-Trans | 3.422 | 0.540 |
| Ranolazine MPO | REINVENT | 3.468 | 0.421 |
| | REINVENT-Trans | **2.727** | **0.434** |
| Scaffold Hop | REINVENT | **2.857** | **0.555** |
| | REINVENT-Trans | 4.355 | 0.382 |
| Sitagliptin MPO | REINVENT | **2.639** | **0.692** |
| | REINVENT-Trans | 5.279 | 0.391 |
| Thiothixene Rediscovery | REINVENT | **2.899** | 0.373 |
| | REINVENT-Trans | 3.275 | **0.441** |
| Troglitazone Rediscovery | REINVENT | **3.275** | **0.441** |
| | REINVENT-Trans | 4.435 | 0.204 |
| Valsartan Smarts | REINVENT | 3.421 | 0.874 |
| | REINVENT-Trans | 3.421 | 0.874 |
| Zaleplon MPO | REINVENT | **1.991** | **0.614** |
| | REINVENT-Trans | 2.465 | 0.486 |

Table 3: Avg SA and Diversity Top100

| Model | SMILES | Score | Number |
|---|---|---|---|
| REINVENT-Transformer | Cc1csc(NC(=O)c2ccc(N3CCCC3=O)cc2)n1 | 0.9479 | 1656 |
| REINVENT-Transformer | COc1cc(NC(=O)c2cnn(C)c2)cc(Cl)c1Cl | 0.9477 | 1875 |
| REINVENT-Transformer | Cc1ncsc1CNC(=O)c1cc(C(F)(F)F)cn1C | 0.9475 | 1873 |
| REINVENT-Transformer | Cc1cc(C(F)(F)F)nn1CC(=O)Nc1ccc(C#N)cc1 | 0.9474 | 466 |
| REINVENT | CS(=O)(=S)c1ccc(C(=O)Nc2ccc(F)cc2)cc1 | 0.9481 | 6853 |
| REINVENT | Cc1ccc(C(=O)Nc2c(F)cc(F)cc2C(=O)N(C)C)o1 | 0.9481 | 5825 |
| REINVENT | Cc1ccc(C(=O)Nc2ccc(S(C)(=O)=O)c(F)c2)cc1 | 0.9481 | 4525 |
| REINVENT | Cc1ccc(S(C)(=O)=O)cc1C(=O)Nc1ccc(F)cc1 | 0.9481 | 4605 |

Table 4: Randomly selected SMILES generated by the REINVENT and REINVENT-Transformer Models

1. **Versus REINVENT (SMILES and SELFIES):** REINVENT-Trans has outperformed the REINVENT model (using SMILES) in multiple instances. However, it's worth noting that in some oracles like 'Osimertinib_MPO', REINVENT achieves a marginally better score. It's also evident that SELFIES representation in REINVENT doesn't always improve the performance as compared to its SMILES counterpart. This underscores the importance of the underlying model's architecture and how different representations can influence its performance.

2. **Graph-based Models:** Both 'Graph GA' and 'GP BO' exhibit competitive performance in certain oracles like 'Amlodipine_MPO' and 'Celecoxib_Rediscovery' respectively. However, their performance isn't

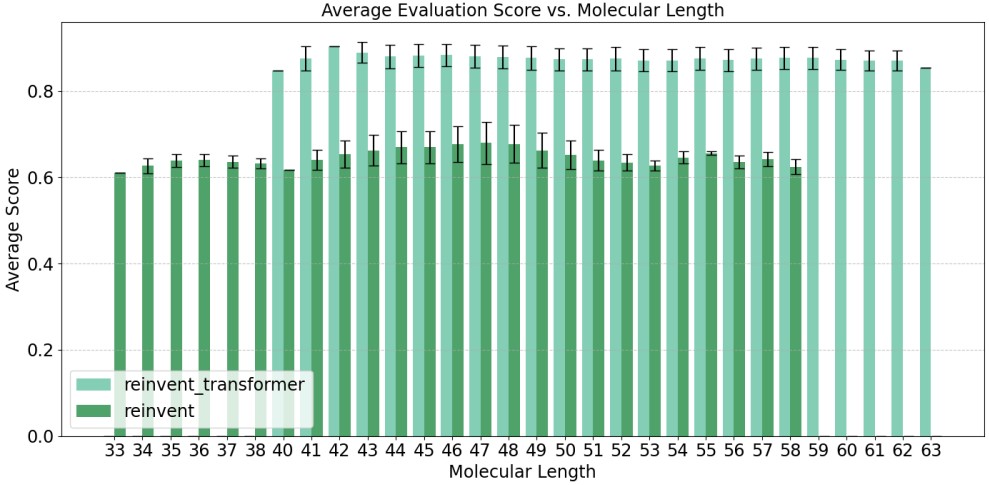

Figure 2: Randomly selected SMILES chemical structures generated by the different models

Figure 3: Evaluation score vs molecular length for comparison of REINVENT-Transformer and REINVENT on oracle Mestranol_Similarity

consistently at the top across all oracles. This implies that while graph-based models can be effective in certain scenarios, they may not always generalize well across diverse tasks.

3. **Genetic Algorithms:** STONED (using SELFIES representation) achieves the highest score in the 'Fexofenadine_MPO' oracle. Genetic algorithms, despite their inherent stochasticity, have potential in some specific optimization tasks.

**Ablation Study: Long Sequence Molecule Generation Comparison with REINVENT SMILES**

In order to better investigate in the ability of our method in long sequence generation, we did the following ablation study.

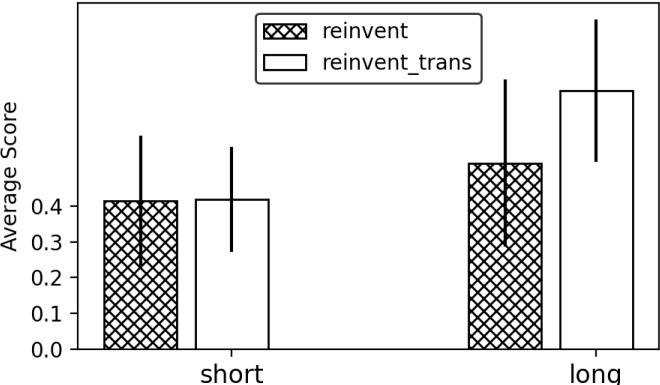

Figure 4: Evaluation score vs short and long sequence for comparison of REINVENT-Transformer and REINVENT on oracle Mestranol_Similarity

The box plot visualizes the distribution of evaluation scores across different molecular lengths for both the REINVENT-Transformer method and the baseline REINVENT method.

Based on the figure 3, we can derive the following observations:

1. In general, REINVENT-Transformer will generate longer average length of molecules than REINVENT.

2. The REINVENT-Transformer method consistently achieves higher average scores.

3. The spread (interquartile range) of scores for the REINVENT-Transformer method remains relatively consistent across molecular lengths, indicating stable performance.

In conclusion, the REINVENT-Transformer method outperforms the baseline REINVENT method, particularly in the context of longer molecular sequences.

We set a threshold=50 for the length of generated molecular string. If the generated string is longer than the threshold, it will be considered as "long", other it's considered as "short" . From the Figure 4, we can see the our method REINVENT-Transformer has better average score when generating long sequences.

**Case Study: Convergence rate Comparison between REINVENT-Transformer and REINVENT**

We plotted the auc_topk curve and number of epoches is the x-axis. From the figure as follows, we can see that our method REINVENT-Transformer converges faster than REINVENT method.

From Fig. 5, the evolution of the average accuracy for the top 100 predictions is evident. Upon examination, across equivalent epochs, the mean accuracy of REINVENT-Transformer consistently surpasses that of REINVENT. This indicates a more expedient convergence rate for the REINVENT-Transformer compared to REINVENT. The avg_top100 curve initially displays a steep incline, eventually plateauing post approximately 6000 epochs. Notably, beginning from the 2500th epoch, the performance differential between REINVENT-Transformer and REINVENT significantly widens.

It is also observed that the REINVENT-Transformer possesses a higher standard deviation relative to REINVENT, suggesting potential variability in its performance. Despite this, the difference between the average top100 accuracy and the standard deviation for REINVENT-Transformer remains superior to the mean accuracy of REINVENT, reaffirming the enhanced efficacy of the REINVENT-Transformer method.

Furthermore, the AUC top100 curve for Albuterol Similarity is illustrated in Fig. 6. In this context, the differential in performance between REINVENT-Transformer and REINVENT is more nuanced. It isn't

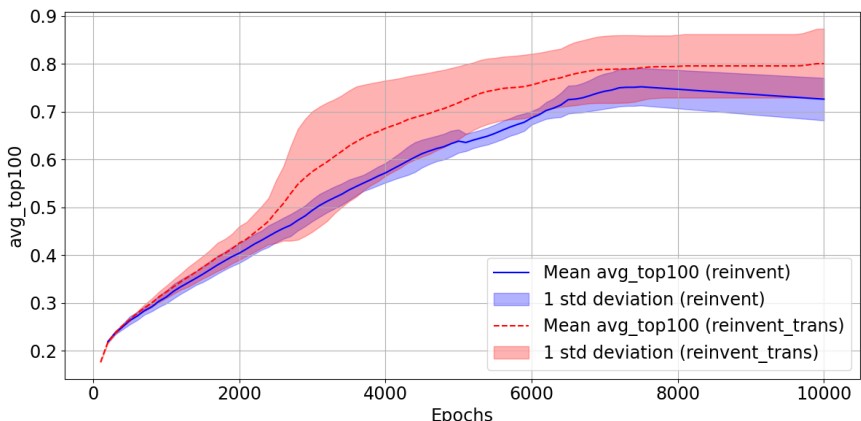

Figure 5: Mean and Standard Deviation of avg_top100 over Epochs for REINVENT and REINVENT-Transformer on oracle Mestranol_Similarity

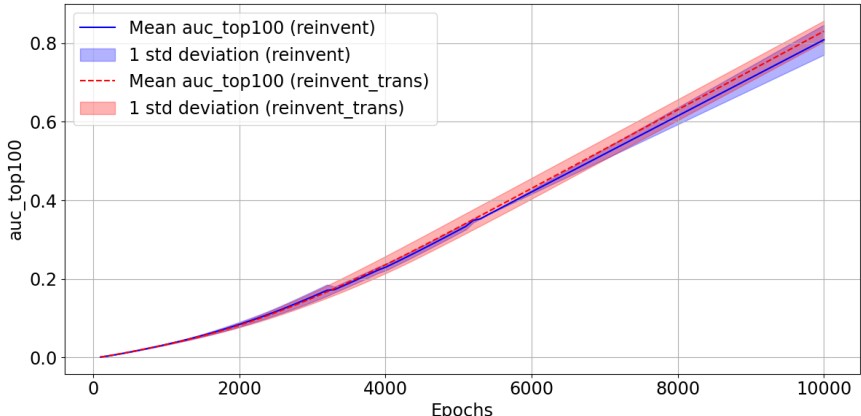

Figure 6: Mean and Standard Deviation of auc_top100 over Epochs for REINVENT and REINVENT-Transformer on oracle Albuterol_Similarity

until the 8000th epoch that a discernible gap emerges. Ultimately, the REINVENT-Transformer exhibits marginally superior performance relative to REINVENT in this scenario.

In Fig. 7, the AUC top10 curve for Mestranol Similarity is presented. Contrasted with the average accuracy curve, this AUC curve demonstrates a milder inclination initially, followed by a pronounced rise. Specifically, for the REINVENT-Transformer, the mean AUC top10 consistently surpasses that of REINVENT. Although the disparity is subtle during the initial epochs, it becomes more pronounced post the 5000th epoch and remains so thereafter.

## Conclusion

Navigating the vast chemical space in molecular design remains a challenge. The introduction of the REINVENT-Transformer marks a significant advancement, harnessing the Transformer architecture's strengths such as parallelization and long-term dependency handling. Our experimental findings reinforce the REINVENT-Transformer's superior performance across multiple oracles, especially in tasks requiring longer sequence data. By integrating oracle feedback reinforcement learning, our approach achieves heightened precision, favorably impacting drug discovery efforts. In essence, the REINVENT-Transformer not only

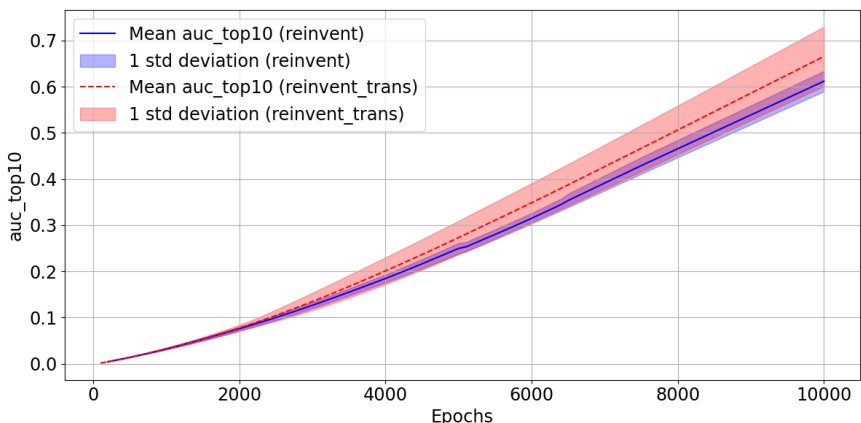

Figure 7: Mean and Standard Deviation of auc_top10 over Epochs for REINVENT and REINVENT-Transformer on oracle Mestranol_Similarity

sets a benchmark in molecular de novo design but also illuminates the path for future research, highlighting the promise of Transformer-based architectures in drug discovery.

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
