# OpenReview forum: "Molecular De Novo Design through Transformer-based Reinforcement Learning"
_TMLR — Rejected by TMLR_

### Review · Reviewer_EMJC · 2023-11-13

**Summary Of Contributions:**

This paper considers the problem of molecule generation, with a particular focus on conditional generation with respect to some set of desired molecular properties.  They propose to pre-train a transformer model that will perform molecular generation, and to fine-tune this model using reinforcement learning to guide the generation process so that successively sampled molecules hew closely to the desired molecular properties.

**Audience:**

No

**Broader Impact Concerns:**

No broader impact statement is warranted.

**Claims And Evidence:**

No

**Requested Changes:**

In general, the writing is disjointed, and the sections are unclear.  But here are some mandatory suggestions for the authors' consideration:

- The final sentence in the related works section is too disconnected from the list of methods in the Related works section.  Which developments are the authors building on specifically?  And what is their analysis of the strengths & weaknesses of prior work that has lead them to propose a transformer and RL as a training regime?
- In the methodology section, Goodfellow  2016 is a textbook of DL, that makes no mention of transformers.  Please cite (at least) Vaswani et al 2017.
- The baseline subsection of the Experiment section is oddly formatted.  This listing of methods would be better formatted as a table to make it more compact and digestible.  As it is, it spans three pages and does not have a single set of criteria to contrast all the baselines.
- The **sample efficiency** subsection describes indirectly the use of 'oracle calls'. From this usage, I suspect what the authors mean here by an ‘oracle call’ is a generated sample.  Is this so?  Why not simply use that term?  If not, please explain what the term means.
- Both tables 1 and 2 are ill formatted.  The amount of information conveyed is little, and they both occupy a lot of space.  Please consider what each is meant to convey, and reformat them compactly.

**Strengths And Weaknesses:**

This paper is likely written, if not entirely generated, by an LLM. This is in contravention to the TMLR policy of the use of LLMs for helping with submissions, where the use of such tools must be explicitly stated.  All the citations in the first two sections (Introduction and Related Works) are to non-sequitur papers from ~50 years ago that have no bearing on the paper's subject matter. In addition, where there should be obvious canonical references (e.g for the transformer architecture, I'd expect Vaswani et al 2017), there is instead this:

> Despite these advancements, challenges such as capturing long-term dependencies in the sequence data persist. The Transformer architecture (NASA, 2015), known for its self-attention mechanism and ability to handle long sequences, has been highly successful in several sequence prediction tasks across domains. Motivated by these successes, we propose the use of Transformer-based architectures in place of RNNs for molecular de novo design.

The (NASA, 2015) reference is to a webpage about the celestial body Pluto. There are several other examples of nonsense citations, which make me suspect the paper was written by an LLM.  LLMs are not forbidden as writing tools per the JMLR faq, but the authors do not state that they have used such a tool, as the submission policy requires.

A brief summary of the weaknesses of this paper.  Try as I might, I found no strengths to praise:
- There is no genuine methods section.  None of the architecture of the transformer model, its pre-training objective, the RL objective, or the RL fine-tuning procedure are described in sufficient detail, and thus cannot be critically evaluated.
- There are several references to papers that are complete non-sequiturs.  For instance, Clancey 1983 is supposed to inform the reader about RNNs and RL for molecular optimization, but the paper is concerned with neither of those subjects. The transformer architecture (whose canonical citation is Vaswani et al 2017) is omitted in favour of a website from NASA pertaining to Pluto.  I do not know where to begin criticizing this.
- The method should be identified by name at the beginning of the methodology section. The reader would be better served by having the subsections summarized in order here in the opening paragraph, to prepare their understanding of the method.
- The paper's writing is far too superficial and vague when describing the details of various models.  For example, in the **Transformers Overview** section, they are described as
> The transformer’s ability to selectively focus on the parts of the input sequence that are most relevant for each step makes them especially well suited for tasks in the field of natural language processing.

This might be true, but has no bearing on the problems of this paper, which is on chemical structures that do not share features of a generative model of text.  It's also not always true of transformers, nor is it exclusively true of this model class.  Furthermore, in the model section, there is no need for a cursory overview of transformers.  Instead, please tell us what specific type of transformer is used here; what are the inputs?  What is the output?  What do the transformer blocks look like, and how is it trained?

- The subsection **Learning the data** is strangely entitled. Training your model is not ‘learning the data’, but rather learning to model the data.
- This same section has an incorrect description of back-propagation, and a non-sequitur description of the phenomena of vanishing gradients and transformers.  It is very odd.
- The training data would be better described concurrently with the methodology section, as aspects of the data are relevant and referenced therein. For example, the SMILES coding describes the number of unique tokens in the training set before the training set is itself described.
- The training set does not have a citation of its own, which is odd, only a footnote pointing to a Kaggle competition link. I suspect even the competition organizers have a preferred way to cite the dataset.
- The comparison metric of the evaluation sections is confusingly described. The "AUC of the top-K average property value in relation to the number of oracle calls" should be defined formally so the reader can understand what is being evaluated.  In particular, what does ‘in relation to the number of oracle calls’ mean here?  How does a model use an oracle call at inference time?
- In the Evaluation results section, the authors state that REINVENT-Trans (presumably the method proposed here, it is not explicitly stated)
> demonstrates its strength in molecular generation, consistently achieving top results in several oracles

What the authors refer to here as "oracles" would more accurately be described as "properties" or possibly “molecular categories”.  The term ‘oracle’ has a more specific term in machine learning literature, usually as an extraneous  (and unmodeled) generator of labels or properties.  The authors’ usage is inconsistent with that understanding, and so I think they should avoid using the term here.

- Tables 1 and 2 are meant to summarize and present the evidence establishing the performance of different methods for molecular generation, but there is little organization in the results, and reading these it is difficult to imagine what to conclude from this.  The authors method is sometimes better, other times less so.  The authors would do well to imagine how to present their results so as to communicate a claim to the readers, and to justify & convince them of it.
- Figure 2 is a comparison of Reinvent (presumably without a transformer, but this is not described) with Reinvent Transformer, but neglects to produce results for the latter in the 'molecular length' of 33-38 and 59-63.  Presumably this is the number of represented atoms in the SMILES string outputs?  Why do these not have the same support?
- The ablation study references a figure without naming it
- The REINVENT method, which is not described, should be.

---

> ### Author Response · Authors · 2024-01-09
>
> Thank you for your review. We have fixed the citation issues like The (NASA, 2015).
>
> - There is no genuine methods section.       Now we have detailed introduction in the method section about the transformer we use in mathematical formula, and how the reinforcement learning is applied.
> - There are several references to papers that are complete non-sequiturs.      Now the problem of citation is fixed.
> - The method should be identified by name at the beginning of the methodology section.  Thank you for notifying, now we identify the method by REINVENT-Transformer.
> - The paper's writing is far too superficial and vague when describing the details of various models.  Now we added mathematical formula for transformer we used. We also added the algorithm of pretraining and finetuning stages.
> - The subsection **Learning the data.** Now it’s changed into ‘learning to model the data’.
> - The training data would be better described concurrently with the methodology section, as aspects of the data are relevant and referenced therein. For example, the SMILES coding describes the number of unique tokens in the training set before the training set is itself described.               We added th training data in the “Transformer-based Molecular Pre-training” before the overview of transformer.
> - The training set does not have a citation of its own, which is odd, only a footnote pointing to a Kaggle competition link. I suspect even the competition organizers have a preferred way to cite the dataset.                Thank you for pointing out, now we change it into ZINC citation.
> -
> - This same section has an incorrect description of back-propagation, and a non-sequitur description of the phenomena of vanishing gradients and transformers. It is very odd.                       Now we fix the description of back-propagation, and remove the non-sequitur  content of vanishing gradient
> - The comparison metric of the evaluation sections is confusingly described. The "AUC of the top-K average property value in relation to the number of oracle calls" should be defined formally so the reader can understand what is being evaluated. In particular, what does ‘in relation to the number of oracle calls’ mean here? How does a model use an oracle call at inference time?                                                                                                                            Now we make this metric section clearer.
> - In the Evaluation results section, the authors state that REINVENT-Trans (presumably the method proposed here, it is not explicitly stated)
>
>     > demonstrates its strength in molecular generation, consistently achieving top results in several oracles
>     >
>
> What the authors refer to here as "oracles" would more accurately be described as "properties" or possibly “molecular categories”. The term ‘oracle’ has a more specific term in machine learning literature, usually as an extraneous (and unmodeled) generator of labels or properties. The authors’ usage is inconsistent with that understanding, and so I think they should avoid using the term here.
>
> Thank you for your suggestion, our usage of “oracle” is mainly borrowed from the paper “Sample Efficiency Matters: A Benchmark for Practical Molecular Optimization”
>
> - Tables 1 and 2 are meant to summarize and present the evidence establishing the performance of different methods for molecular generation, but there is little organization in the results, and reading these it is difficult to imagine what to conclude from this. The authors method is sometimes better, other times less so. The authors would do well to imagine how to present their results so as to communicate a claim to the readers, and to justify & convince them of it.
>
>     The format of table1 is from“Sample Efficiency Matters: A Benchmark for Practical Molecular Optimization”, which shows each baseline methods in this large table.    As we can see in the benchmark paper, it’s often the case that one algorithm has superior performance in one task but inferior in another.   The sum of AUC each method shows their performance in general.

---

> ### Author Response · Authors · 2024-01-09
>
> For Request Changes:
> - The final sentence in the related works section is too disconnected from the list of methods in the Related works section. Which developments are the authors building on specifically? And what is their analysis of the strengths & weaknesses of prior work that has lead them to propose a transformer and RL as a training regime?
> - In the methodology section, Goodfellow 2016 is a textbook of DL, that makes no mention of transformers. Please cite (at least) Vaswani et al 2017.                            Now it changed into correct citation of transformer.
> - The baseline subsection of the Experiment section is oddly formatted. This listing of methods would be better formatted as a table to make it more compact and digestible. As it is, it spans three pages and does not have a single set of criteria to contrast all the baselines.                                              Thank you for your notification, this listing of methods now is converted to a table.
> - The **sample efficiency** subsection describes indirectly the use of 'oracle calls'. From this usage, I suspect what the authors mean here by an ‘oracle call’ is a generated sample. Is this so? Why not simply use that term? If not, please explain what the term means.                                  oracle definitation is now in the first part of the method.
> - Both tables 1 and 2 are ill formatted. The amount of information conveyed is little, and they both occupy a lot of space. Please consider what each is meant to convey, and reformat them compactly.                                   now it’s more compact.

---

### Review · Reviewer_kmr5 · 2023-12-05

**Summary Of Contributions:**

This paper proposes a molecular generation approach which utilizes transformer models to generate SMILES strings and reinforcement learning to optimize desired objectives. The method is developed based on a molecular generation algorithm named REINVENT and replaces it RNN-based SMILES generator to transformers. The proposed method achieves SOTA performance on molecular generation benchmark. However, it has limited contributions in model architecture and inaccurate conclusions. Major revisions are also required for the writing.

**Audience:**

Yes

**Claims And Evidence:**

No

**Requested Changes:**

1. Provide more details regarding the proposed method
2. Correct the figures and statements listed by Weakness item 3.
3. Correct writing related problems listed by Weakness item 4.

**Strengths And Weaknesses:**

Strengths:
1. The proposed method achieves SOTA performance on molecular generation benchmark.
2. The authors proves that transformers is more suitable than RNNs in REINVENT architecture under some evaluation metrics.

Weaknesses:
1. Lack of contributions in methodology: the proposed method simply replaces the RNNs in REINVENT to transformers.
2. Missing method details:  In Methodology, the authors describe general concepts such as transformer architecture, back propagation, on-policy and off-policy learning, value-based and policy-based RL, SMILES strings, etc. Some of these concepts are not directly related to the proposed method and can be included in the supporting information rather than the main text. In addition, limited details related the proposed method are provided, including loss functions, how the policy is formulated in RL and etc.
3. Inaccurate statements and figures:
    1. In Related Works, the authors mentions structure-based de novo design tends to generate molecules with poor DMPK properties without providing any references. Actually ligand-based and structure-based drug design differs by whether the target 3D structure is used in the design process. [1] There is no guarantee that ligand-based approaches are superior than structure-based ones.
    2. In Ablation Study Figure 2, REINVENT-Transformer’s results of molecular length between 33 and 38 are missing and REINVENT’s results of molecular length between 59 and 63 are missing. In the corresponding text, it is stated that both methods exhibit similar score distribution for shorter molecular lengths, and when molecular length gets longer the differences in scores become more significant. This conclusion cannot be inferred from the Figure 2.
4. Major revisions are necessary for the writing.
    1. No definition of DMPK.
    2. Mixed usage of “transformer” and “Transformer”; “REINVENT”, “Reinvent” and “reinvent” throughout the paper.
    3. REINVENT-Transformer seems to be the name of the proposed model, which is not clarified clearly.
    4. Missing references:
        1. Missing references for Screening, MolPal, VAE, LSTM.
        2. Figure 2 is not referenced in the text.
    5. Formatting:
        1. In Segmentation and Binary Coding of SMILES: an unnecessary line break after “SMILES are tokenized”
        2. Missing line break before “Overall Molecular Generation Result Performance Overview”.
        3. The first 4 paragraphs in Ablation Study have repeated contents.

 [1] Blundell, Tom L. "Structure-based drug design." *Nature* 384.6604 (1996): 23.

---

> ### Author Response · Authors · 2024-01-09
>
> For Request Changes:
>
> 1.Thank you for notifying, now we added more details about the method, like loss function, scoring function.
>
> 2.
>
> weakness item 3.1 cited[1], and clarified that the effectiveness of ligand-based methods compared to structure-based ones is not definitive.
>
> weakness item 3.2 Now we rewrite this section.
>
> 3.
>
> Weakness item 4.1  DMPK is drug metabolism and pharmacokinetic. We added the full name before the abbreviation.
>
> Weakness item 4.2 fixed, now all the names are changed into “REINVENT” and “REINVENT-Transformer”
>
> 4.4.1 all reference mentioned are added. 4.4.2 Figure 2 now referenced in ablation study
>
> 4.5.1 4.5.2 4.5.3 fixed

---

### Review · Reviewer_NPiN · 2023-12-10

**Summary Of Contributions:**

This submission aims to improve the molecular De Novo design based on the generate models. Specifically, the authors replaced RNNs used in previous models with transformers for the pre-training step, with the hope of more representation power. Furthermore, they employed “oracle feedback reinforcement learning” in the fine-tuning step to incorporate oracle feedback.

**Audience:**

Yes

**Claims And Evidence:**

Yes

**Requested Changes:**

1. The authors need to share more details on how RL works in the fine-tuning stage. For instance, what were obstacles to apply RL and how did you overcome them?
2. There are a few confusing or incorrect statements the authors may need to clarify
    1. It was mentioned in p1 that `Transformers process all tokens in the sequence simultaneously, allowing for better computational efficiency`. However, the authors stated that in p4 that `The sampled $x_t$ is then used as our next input`. This is contradictory with each other.
    2. Design of rewards in RL is unclear, without which I cannot access the correctness of this part.
    3. Definition of AUC in p7 is also confusing: how did you define the corresponding recall and precision, based on oracles?
3. For the experiments, could you also provide some qualitative results? I don’t have any background in molecular design so it’s worth consulting some experts in this domain.

**Strengths And Weaknesses:**

Strengths
1. The authors aimed to apply the generative models in an important problem. Use of the transformer over RNNs looks to be interesting and reasonable.
2. Combination with reinforcement learning to incorporate feedback could be significant, though it’s difficult to justify based on the this draft (more details shared below).

Weakness
1. The major issue is the lacking of details for the reinforcement learning part. From p4 to p5, the authors mostly provided well-known basic for RL, but failed to provide enough technical details on how it actually works. For instance, what’s the definition of score function and how is it designed?
2. The rigor in this draft needs to be improved as some statements are either incorrect or confusing.
3. The experimental results may not be sufficiently significant, compared with some other methods.

---

> ### Author Response · Authors · 2024-01-09
>
> 1. We have added the detailed pretrain and finetune stage algorithm pesudocode and the loss function.
>
> 2.1.
>
> 1. **"Transformers process all tokens in the sequence simultaneously"**: This statement refers to the architecture of transformer models, like those used in various GPT versions. In transformers, unlike earlier models like RNNs or LSTMs, each token in a sequence is processed in parallel during training and inference. This parallel processing allows transformers to be more computationally efficient, as they don't require sequential processing of each token, which can be time-consuming.
> 2. **"The sampled $x_t$ is then used as our next input"**: This statement seems to be describing a step in a sequential process, possibly related to sampling methods used in language generation or in some iterative process in the model. It doesn't necessarily contradict the first statement. In the context of language generation, for instance, a transformer model can generate a sequence token by token. For each token generated, the model samples a new token ($x_t$), which is then added to the sequence and used as part of the input for generating the next token. This sequential generation is part of the inference process and does not contradict the parallel processing nature of transformers.
>
> 2.2 In the method section, we’ve added the details of reinforcement learning , including NLL loss and scoring function.
>
> 2.3
>
> 1. **Recall (Sensitivity)**: Traditionally, recall measures the proportion of actual positives correctly identified. In this context, recall could be interpreted as the proportion of molecules with desirable properties (as judged by the oracle) that the method successfully identifies. For example, if there are 'N' molecules with desirable properties according to the oracle, recall would assess how many of these 'N' molecules are correctly identified by the method.
> 2. **Precision (Positive Predictive Value)**: Precision is traditionally the proportion of predicted positives that are true positives. In this scenario, it could be the proportion of molecules identified by the method as having desirable properties that are confirmed by the oracle to possess these properties. If the method selects 'M' molecules as having desirable properties, precision would measure how many of these 'M' molecules are actually validated by the oracle.
>
> 3.Thank you for your notifying. We have included some randomly selected SMILES generated by the REINVENT and REINVENT-Transformer Models in table 4 and figure2.

---

### Decision · Action_Editor_XoX5 · 2024-01-28

**Recommendation:** Reject

**Comment:**

The paper was reviewed by three reviewers. Unfortunately, after rebuttal, all three reviewers are negative and recommend reject or leaning reject. The main concerns include lack of important details and unsubstantiated conclusions. One reviewer also raised the concern that the authors might have relied on LLM in writing up the paper.

Specifically, a reviewer thinks the revised version is still not written in a way that will be useful to the ML community. For example, even in the revised version, it is not clear why the authors propose to use oracle based RL to fine-tune their pre-trained transformer model. In contrast to the paper that the authors cite as their most direct antecedent (Olivecrona et al), which describes the development of how to specify the complexity of the objective for de-novo molecular generation models, and which motivates their use of RL to learn a more nuance reward function, the authors here simply allude to using RL and that it's better. A reader would quite rightly want to know why, what are the alternatives, and why should RL be employed (and when?). While some of this is hinted at in the related works section (there's a reference to Jacques et al), there's no connection of the concepts in this paper.

Furthermore, this reviewer stresses that the updated methods section (pages 6-9 of the revised version) are simply not described in a level of detail that would be useful to someone that wished to build upon this work. Pages 6 and 7 are whole-page pseudo-code blocks that attempt to describe the pre-training of a transformer on prior chemical datasets. Pages 8 and 9 describe an introduction to RL in general, but not rather how the principles are used for this application (which is critical).

Finally, the reviewer notices that some of the references seem invented out of whole cloth. The updated reference for the Transformer architecture is now Vaswani et al, however they did not publish the work in 2023, as claimed in the paper. Another cursory example is that the reference to Jacques et al 2017 includes two authors who are not on the paper, and does not provide a publication venue (it was a workshop paper at ICLR 2017). This lies somewhere between poor scholarship, and using LLMs (uncredited, again in contravention of the TMLR policy) for writing the manuscript.

**Audience:**

The paper should be interesting to readers who are interested in AI for science in general, and AI for drug design in particular.

**Claims And Evidence:**

This paper proposes a transformer-based reinforcement learning method for molecule design. The transformer-based method achieves superior performance compared to previous method based on recurrent neural networks. The claims made in the submission are partially supported by experimental evidence. However, as pointed out by one reviewer, some conclusions are not supported by the presented results.